# An In Vitro Protocol to Study the Modulatory Effects of a Food or Biocompound on Human Gut Microbiome and Metabolome

**DOI:** 10.3390/foods10123020

**Published:** 2021-12-05

**Authors:** Carles Rosés, Juan Antonio Nieto, Blanca Viadel, Elisa Gallego, Ana Romo-Hualde, Sergio Streitenberger, Fermín I. Milagro, Anna Barceló

**Affiliations:** 1Servei de Genòmica, Universitat Autònoma de Barcelona, Bellaterra, 08193 Cerdanyola del Vallés, Spain; carles.roses@uab.cat; 2Ainia, In Vitro Digestion Assays, Parque Tecnológico de Valencia, 46980 Paterna, Spain; jnieto@ainia.es (J.A.N.); bviadel@ainia.es (B.V.); egallego@ainia.es (E.G.); 3Centre for Nutrition Research, Department of Nutrition, Food Sciences and Physiology, University of Navarra, 31008 Pamplona, Spain; aromo@unav.es (A.R.-H.); fmilagro@unav.es (F.I.M.); 4AMC Innova, 30100 Murcia, Spain; sstreiten@amunoz.com; 5Navarra Institute for Health Research (IdISNA), 31008 Pamplona, Spain; 6Centro de Investigación Biomédica en Red de la Fisiopatología de la Obesidad y Nutrición (CIBERobn), Instituto de Salud Carlos III, 28029 Madrid, Spain

**Keywords:** microbiota, digestor, food compounds, colonic fermentation, metabolome

## Abstract

The gut microbiota plays a key role in gastrointestinal immune and metabolic functions and is influenced by dietary composition. An in vitro protocol simulating the physiological conditions of the digestive system helps to study the effects of foods/biocompounds on gut microbiome and metabolome. The Dynamic-Colonic Gastrointestinal Digester consists of five interconnected compartments, double jacket vessels that simulate the physiological conditions of the stomach, the small intestine and the three colonic sections, which are the ascending colon, transverse colon and descending colon. Human faeces are required to reproduce the conditions and culture medium of the human colon, allowing the growth of the intestinal microbiota. After a stabilization period of 12 days, a food/biocompound can be introduced to study its modulatory effects during the next 14 days (treatment period). At the end of the stabilization and treatment period, samples taken from the colon compartments are analysed. The 16S rRNA gene analysis reveals the microbiota composition. The untargeted metabolomics analysis gives more than 10,000 features (metabolites/compounds). The present protocol allows in vitro testing of the modulatory effects of foods or biocompounds on gut microbiota composition and metabolic activity.

## 1. Introduction

The outcome of the ingested food components in the human digestive system is an area attracting interest among researchers because of its relation to nutrition and health. Digestion of food is a complex combination of various physiochemical processes that disintegrate the food into more suitable forms for its absorption and transportation to related organs, and discarding the remaining waste [1]. In this context, the gut microbiota plays a key role in the host’s health by metabolizing dietary compounds such as fibres and polyphenols that have a potential prebiotic capacity and is a determinant factor in the development of obesity and other diseases [2]. The diet composition determines the gut microbiota profile, and therefore the diet–microbiota interaction is crucial for synthesizing vitamins and other beneficial bioactive molecules, such as postbiotics [3]. The microbiota also plays a key role in the maintenance of the intestinal functions, modulating the immunological response and working as a barrier against certain pathogens [4]. The composition of our microbiota is influenced by host genotype, environment, and diet. A gut microbiota in a eubiotic status is characterized by a preponderance of potentially beneficial species, belonging mainly to the two bacterial phylum, Firmicutes and Bacteroidetes [3], while potentially pathogenic species, such as those belonging to the phylum Proteobacteria (i.e., Enterobacteriaceae) are present, but in a very low percentage. In case of dysbiosis “good bacteria” no longer control the “bad bacteria” which take over [5]. The composition of the intestinal microbiota and the metabolic activity can be modulated through diet, which opens an opportunity for intervention with dietary strategies to balance the intestinal microbiota and reverse the state of dysbiosis. Additionally, the characterization of new beneficial molecules that are generated by the microorganisms during the colonic fermentation of foods, which are called postbiotics [6]. Increased interest in modifying the matrix and structural characteristics of foods to optimize their digestion, absorption and particularly the microbiota profile (related to health benefits), requires implementation of many food digestion studies in the digestive tract; therefore, it is essential to design an in vitro protocol for analysing the effect of foods and dietary bioactive compounds on the microbiota profile and metabolite production.

The main objective of the present study was to standardize a protocol that allows testing, in vitro, of the modulatory effects of foods or biocompounds on gut microbiota composition and metabolic activity.

## 2. Material and Methods

### 2.1. In Vitro Digestion Models

In vitro models have been developed since the 1990s to be used in food digestion studies. The models can help to design novel food products by estimating the in vivo behaviour after meals [7]. Many of these models are designed to work in static conditions, and therefore cannot reproduce the dynamic conditions that occur in the digestive system. Furthermore, these models have limitations for predicting food behaviour and nutrient bioavailability [8]. The interest that has arisen in the last years about the gut microbiota physiology has obliged to design new in vitro models more centred on colonic fermentation. In this context, very few dynamic in vitro models mimic the mechanic, dynamic and chemical conditions of the human digestive system. Additionally, multi-compartmental systems usually have gastric and small intestinal compartments, but very few include the colon, which is essential to study of the gut microbiome and metabolome. Dynamic models can simulate the change in pH, enzyme secretion, peristaltic forces and microbial fermentation continuously [9].

### 2.2. Description of the Dynamic Gastrointestinal and Colonic Fermentation Model 

The equipment simulates in vitro the entire gastrointestinal digestive process. It consists of a computer-assisted model of five interconnected compartments, double jacket vessels, which simulate the physiological conditions of the stomach (R1), the small intestine (R2) and the three colonic sections: the ascending colon (R3), transverse colon (R4) and descending colon (R5). This model was developed by AINIA (Valencia, Spain) [10] based on the work of Van de Wiele et al. [11] and Marzorati et al. [12]. R1 and R2 work semi-continuously, while the colon reactors (R3, R4 and R5) work continuously. A peristaltic bomb ensured the flow of the content from one reactor to the next. The system did not simulate water absorption.

The volumes and transit times for each region of the gastrointestinal tract were: 260 mL for 2 h in R1, 410 mL for 6 h in R2 and the colon, 1000 mL for 20 h in R3, 1600 mL for 32 h in R4, and 1200 mL for 24 h in R5. [13,14,15]. The temperature was kept at 37 °C during the entire process. An anaerobic environment was maintained, and gaseous N_2_ was flushed for 15 min twice a day.

Gastric digestion was simulated by continuous addition of a 0.03% (*w*/*v*) pepsin solution (2100 units/mg) during 2 h (total volume of 60 mL). A typical gastric digestion pH curve (based on in vivo data) was simulated by adding a 1 M HCl solution. The gradual decrease in the pH of the stomach to pH 2 was carried out. The pH control was carried out by adding HCl. HCl secretion to control pH is based on the pH curve reported by Conway et al. (1987), which represents the pH profile in the stomach after human volunteers consumed yogurt [16]. Digestion of the small intestine was mimicked by the continuous addition of a pancreatin solution (0.9 g/L), NaHCO3 (12 g/L) and oxgall dehydrated fresh bile (6 g/L) in distilled water (total volume of 440 mL), maintaining the intestinal content at pH 6.5. 

### 2.3. Faecal Inoculum

Fresh faeces from human subjects were used to reproduce the gut conditions. A list of characteristics can be added in the recruitment procedure depending on the aim of the experiment. As the faeces profile determines the characteristics of the microbiota reproduced in the digestor, especially when a dysbiosis or normobiosis environment is the study aim, this parameter should be considered. In this case, the inoculum was prepared using faeces from four adult volunteers with pathologies associated with obesity and/or metabolic syndrome (BMI 35–40; age 30–50), non-smokers, no history of antibiotic treatment in the last three months, and no intestinal disease background [17,18,19].

Faecal samples were collected and maintained in special anaerobic plastic bags (BD GasPak™ systems). Faeces were diluted with thioglycolate 20% (*w*/*v*) and homogenized with a stomacher to obtain a faecal slurry. The faecal suspension was centrifuged at 3000 g for 3 min and the collected supernatant was immediately inoculated in the colon vessels (50, 80 and 60 mL for R3, R4 and R5, respectively). The reactors were filled with culture medium up to a total volume of 1000, 1600 and 1200 mL, respectively. The composition of the culture medium followed Molly et al. (1993, 1994) [20,21], providing the necessary nutritional components to simulate the conditions of the human colon and allowing the intestinal microbiota to grow. Each reactor was maintained at different optimal pH levels. The bacteria present in each region of the colon have an optimal pH of action: pH 5.5–6 in the ascending colon (R3); pH 6–6.4 in the transverse colon (R4) and pH 6.4–6.8 in the descending colon (R5). In order to regulate the pH changes that occur during fermentation and maintain them in the optimal intervals for each region, acid or base were added.

### 2.4. Process Description and Duration

After faecal inoculation, a stabilization period of 12 days was required to allow bacteria to grow and reach stable levels [22,23]. During this period, 200 mL of cultured medium was added to the stomach (R1) three times a day. At this point, the system was ready to start the sample treatment period to study its modulation effects on the gut microbiota [10,11]. The treatment period was conducted by feeding the equipment with sample once a day (in culture medium up to 200 mL) and with 200 mL of cultured medium twice a day. The treatment period was 14 days. The maintenance of the microbial population during the stabilization (time 12) and treatment period (time 14) was checked by bacteria plate counts. The following bacterial groups were quantified by growth on specific medium, expressing the result as CFU/mL of colonic medium: *Lactobacillus* (MRS agar; the MALDI-TOF technique was employed to verify lactobacilli colonies), *Bifodobacterium* (TOS-propionate agar enriched with MUP), Enterobacteriaceae (VRBD agar), *Clostridium* (TSC Agar enriched with cycloserin) and total anaerobic bacteria (Agar Schaedler). Then, 10 mL of the samples was taken from each reactor (R3, R4 and R5) and serially diluted in saline solution. Plates were inoculated with 1-mL sample of four serial dilutions by duplicate and incubated at 37 °C under aerobic or anaerobic conditions. In addition, samples were taken from each reactor at the end of the stabilization and treatment period and stored at −20 °C to determine the short chain fatty acid content in the colonic media and to conduct the metabolomic analysis, 10 mL and 3 mL, respectively. For the metabolomic analysis, these samples were centrifuged (15,000× *g*, 15 min) and filtered through a 0.22-µm-Ø Millipore filter (Billerica, MA, USA) into vials for UHPLC-ESI-QqQ-MS/MS analysis. In addition, 1.5 mL of the inoculum, as well as samples from each reactor at the end of the stabilization and in the treatment period were collected at baseline using OMNIgene.GUT kits from DNA Genotek (Ottawa, ONT, Canada), according to the standard instructions provided by the company. 

### 2.5. Description and Volume of the Product to Be Tested

A vegetable drink based on oats, fruit, vitamins B2, B5, B12 and D2, iodine, calcium, beta-glucans and postbiotics, with a 6% sugar content, was pasteurized to ensure a viability of, at least, 15 days. It was stored at −20 °C. The dosage administrated during the treatment was about 100 mL/day, once a day. The product was provided by the company AMC Natural Drinks, a partner of the BIOTAGUT Project (Modulation of the microbiome and postbiome by the intelligent design of food promoters of a healthy microbiota in relation to metabolic syndrome), which is supported by the Ministry of Science, Innovation and Universities of Spain, through CDTI (Centre for Industrial Technological Development, Madrid, Spain).

### 2.6. Short Fatty Acid Analyses

Short and medium chain fatty acids were extracted from the sample using a liquid–liquid extraction with diethyl ether. The resulting extract was filtered and subsequently analysed using a AS 800 C.U. gas chromatograph (CE Instruments, Wigan, United Kingdom) equipped with a HP-FFAP 25 m × 0.2 mm × 0.33 mm column (Agilent Technologies) and a flame-ionization detector (FID). The samples were quantified by interpolation in the calibration curve using capric acid as an internal standard. The concentration of the fatty acids was provided directly by the software using a 1/× linear regression. The results were expressed as mg of compound per Kg of colonic medium.

### 2.7. DNA Extraction

The DNA was extracted with the QIAamp^®^ DNA kit (Qiagen. Hilden, Germany) following the manufacturer’s protocol [24].

Microbiota composition according to 16S rRNA analysis with Next Generation Sequencing (NGS).

### 2.8. Metagenomic Data: Library Preparation

Metagenomics studies were performed by analysing the variable regions V3–V4 of the prokaryotic 16S rRNA (ribosomal Ribonucleic Acid) gene sequences, which gives 460 bp amplicons in a two-round PCR protocol.

In a first step, PCR is used to amplify a template out of a DNA sample using specific primers with overhang adapters attached to the flank regions of interest. The full-length primer sequences, using standard IUPAC (International Union of Pure and Ap-plied Chemistry) nucleotide codes, were: Forward Primer: 5′TCGTCGGCAGCGTCAGATGTGTATAAGAGACAGCCT ACGGGNGGCWGCAG; and Reverse Primer: 5′GTCTCGTGGGCTCGGAGATGTG TATAAGAGACAGGACTACHVGGGTATCTAATCC. PCR was performed in a thermal cycler using the following conditions: 95 °C for 3 min, 25 cycles of (95 °C for 30 s, 55 °C for 30 s, and 72 °C for 30 s), and 72 °C for 5 min.

To verify the amplicon, 1 µL of the PCR product was checked in a Bioanalyzer DNA 1000 chip (Agilent Technologies, Santa Clara, CA, USA). The expected size on a Bioanalyzer was ~550 bp.

In a second step and using a limited-cycle PCR, sequencing adapters and dual index barcodes, Nextera^®^ XT DNA Index Kit, FC-131-1002 (Illumina, San Diego, CA, USA), were added to the amplicon, which allows up to 96 libraries pooled together for sequencing in the MiSeq sequencer with the MiSeq^®^ Reagent Kit v2 (500 cycle) MS-102-2003 to be pooled together.

The PCR was performed in a SimpliAmp thermal cycler (Applied Biosystems^®^, San Francisco, CA, USA) using the following conditions: 95 °C for 3 min, eight cycles of (95 °C for 30 s, 55 °C for 30 s, and 72 °C for 30 s), and 72 °C for 5 min. Subsequently, the Index PCR ran a second Bioanalyzer DNA 1000 chip to validate the library. The expected size was ~630 bp.

The next step consisted of quantifying the libraries using a Qubit^®^ 2.0 fluorometer (Invitrogen by Thermo Fisher Scientific, Carlsbad, CA, USA), and dilution of the samples before pooling them.

Finally, paired-end sequencing was performed in a MiSeq platform (Illumina) with a 500-cycle Miseq run and with 7 pM sample and a minimum of 25% PhiX. The mean reads obtained were 164,387. Only samples with more than 40,000 reads were used for further analysis [25].

### 2.9. Data Analysis

For the Short-chain fatty acid (SCFA) production and the bacterial growth analysis, a Mann–Whitney U Test was applied for differences between two groups on a single, ordinal variable with no specific distribution [26]. These analyses were carried out using GraphPad Prism version 6.0c for MAC OS X, GraphPad Software, San Diego, California USA, www.graphpad.com (accessed on 13 September 2021).

### 2.10. Metagenomic Data: Analysis and Processing

The 16S rRNA gene sequences obtained were filtered following the quality criteria of the OTU (operational taxonomic units) processing pipeline LotuS (release 1.58) [23]. This pipeline includes UPARSE (Highly accurate OTU sequences from microbial amplicon reads) de novo sequence clustering and removal of chimeric sequences and phix contaminants for identifying OTUs and their abundance matrix generation [27,28]. The taxonomy was assigned using HITdb (Highly scalable Relational Database), achieving up to species sensitivity. BLAST (Basic Local Alignment Search Tool) was used when HITdb failed to reach a homology higher than 97% [29,30]. Thus, OTUs with a similarity of 97% or more were considered as species by themselves. However, OTUs that did not reach this percentage of similarity were checked and updated using the Basic Local Alignment Search Tool (BLASTn), comparing them with the 16S rRNA gene sequences of the bacteria and archaea database of GenBank of the National Center for Biotechnology Information to obtain an assignment to a species. The sequences in which the BLASTn tool found a new assignment were indicated using the GenBank access number and the percentage of homology following the species name. The abundance matrices were first filtered and then normalized in R/Bioconductor at each classification level: OTU, species, genus, family, order, class, and phylum. This study focused mainly on the species level. Briefly, taxa with less than 10% frequency in our population were removed from the analysis, and a global normalization was performed using the library size as a correcting factor and log2 data transformation [31].

### 2.11. Richness and Evenness

Richness was defined as the total of species. Evenness, defined as the alpha diversity index, was calculated using the Shannon index [32] according to the following formula:H = −sum(Pi ln[Pi])(1)

### 2.12. Untargeted Metabolomics

The chromatographic analysis was performed with a high-resolution liquid chromatogram (HPLC) from Agilent (model 1100). The detector used was a TOF (Time of Flight) Mass Accuracy from Agilent, model 6220. The stationary phase was a chromatographic column, Zorbax SB-C18 (Agilent Technologies) of 150 × 46 mm and 5 μm pore size. The column temperature was kept at 40 °C. The mobile phase was made of milliQ with 0.1% formic acid (canal A) and methanol with 0.1% formic acid (canal B). The system worked with a gradient, screening the entire polarity range from 100% of A until 100% of B. The injected volume was 15 μL.

The TOF detector is made of an electrospray ionized source (ESI). The flow of gas drying was 10 L/min, at a 40 psig pressure and a temperature of 350 °C. The capillary voltage was 4000 V, the fragmentor voltage was 175 V, and the skimmer voltage 65 V. The relation m/z range, for the detection, was between 100 and 2000 and the acquisition ratio was 1.03 spectre/sec.

All samples coming from the digestor were analysed following the same sequence. The samples were analysed in positive polarity (POS group) and negative polarity (NEG group). To apply this process, sample duplicates were needed to perform the two analyses.

To analyse the results, an alignment with XCMS Online (The Scripps Research Institute, La Jolla, CA 92037, United States) software was carried out according to the relation m/z (mass/charge) and the retention time. Raw data were normalized by logarithmic transformation followed by Pareto scaling. Compounds with the same m/z relation and the same retention (5 mDa tolerance for the m/z relation; 0.5 min for the retention time) were considered the same.

Later, comparisons of two vs. two were carried out, for example: R3 T12 POS vs. R3 T14 POS and R3 T12 NEG vs. R3 T14 NEG. The same with the three reactors. These comparisons show the impact of the chosen treatment on the microbiota of the in vitro digestor. The XCMS Online software gave features (metabolites/compounds) detected from each chromatogram, showing more than 10,000 features. Finally, the MetaboAnalyst software (Xia Lab of McGill University, Quebec, Canada) processed the list file using statistical multivariable tools, such as Partial last squares regression (PLS-DA), *Random Forest* and Volcano Plot. Outliers were excluded through the *Random Forest* model, followed by the other statistical tools mentioned above. With these tools, features with more discriminating outputs were chosen between the two groups of the polarity group:-Volcano Plot: *p*-value < 0.001 and Fold change >10.0 (100.0 for negative polarity).-PLSDA: Value of Variable Importance in Projection (VIP) >3.0

Subsequently, the features that passed the selection criteria were identified through the database Metlin (Scripps Research Institute).

Certain conditions must be established to set the tolerance limit: 5 mDa in relation to the m/z ratio (specifically the mean m/z relation of the csv file, generated from the XCMS Online). For the positive group, the marked ions were [M+H]^+^, [M+Na]^+^ and [M+H-H2O]^+^. For the negative group, the ions were [M-H]^−^ and [M-H2O-H]^−^. In both cases, those metabolites classified as toxic were taken out of the database [33].

## 3. Results and Discussion

In the colonic fermentation studies, two different phases exist: the stabilization part, which aims to create a stable microbiome from the faecal inoculum, and is considered as the control, and the treatment one, where the purpose is to analyse the behaviour of the stabilized microbiota under the effects of the chosen food or biocompound, added after the stabilization period is over. During both periods, samples are taken from the colonic digestors, allowing the comparation between the metabolites production by the microbiota itself (stabilization period), and due to the chosen food or biocomppound added (treatment phase).

The test results showed a stabilization of the faecal microbiota during the first period (12 days) in R3, R4 and R5. The recovered levels (for each ml of cultured medium) of total anaerobic microorganisms were above 10^7^ cfu in all three sections. During the treatment period (14 days after reaching the stabilization point), 100 mL of the vegetable drink were added every day. In this period, the total anaerobic microorganisms remained stable in the three sections. According to the total anaerobic level, it was concluded that the digestor conditions were favourable for the development of the intestinal microbiota during the two weeks while the product was being tested. R3 shows a stabilization of the anaerobic bacteria while R4 and R5 show a very significant increase (*p* < 0.025). Furthermore, bacteria from the Enterobacteriaceae family, including potentially pathogenic species, show a very significant (*p* < 0.025) reduction in all reactors. These results are shown in Table 1 and Figure 1.

Short-chain fatty acid (SCFA) production showed a clear tendency towards increased levels of acetic and butyric acids in the three sections after the treatment (Table 2). Propionic acid level showed a very significant (*p* < 0.025) increase after the treatment period in the three reactors (Figure 2).

The metagenomic analysis showed an increase in some bacterial species, (Figure 3), especially butyrate-producer species such as *Alistipes putredinis* and *Eubacterium desmolans.* Other SCFA-related species (*Clostridium lactatifermantans* and *Phascolarctobacterium succinatutens* (propionic)) also increased their levels; these results are shown in Table 3 and Figure 3. Results on the genus rank show a growth of some genera related to SCFA production, such as *Anaerotruncus* [34], *Cloacibacillus* [35], and *Parasutterella* [36]. The genus *Acidaminococcus* [37] has attracted scientific interest due to its high resistance to antibiotics. These results are shown in Table 4.

A total of 219 bacterial species were obtained in this experiment. The result of alpha diversity assessed using the Shannon index showed an increase at R3 with the treatment, however not at R4 and R5, where it decreases (Table 5)

The metabolomic analysis showed a dominance of peptides consisting of up to four amino acids, phenolic compounds, phosphatidylcholines, fatty acids and terpenoids (Table 6). For example, in negative polarity, several interesting metabolites increased their levels. In R3, there are some catechin derivatives and an isoflavone; in R4, a metabolite related to vitamin D3; in R5, beta-carotenes or lycopene; and in R3 and R5, there is a terpenoid. In positive polarity, there is an increase in choline in R4. Figure 4 shows the PLS-DA Score Plots obtained in the comparisons among all groups in positive (A) and in negative (B) polarity.

With this model, mechanic, dynamic and chemical conditions of the human digestive system can be outlined, including transit times of ingested meals, pH profiles, temperature, contractions, and peristaltic movements, digestive secretion rates, and absorption of water and nutrients. Moreover, it also allows analysis of the microbiota fluctuation and the metabolites produced along the whole process. However, as it is mainly designed for studying the effects of foods on microbiota composition and gut metabolites, it does not take into account nutrient absorption and gut leak. The main differences of the intestinal fermentation protocol established in this article compared to other similar protocols (SHIME, TIM-2, SIMGI,…) previously published [1,38] are: 

The system can adapt the gastrointestinal transits and the addition of digestive secretions to the state of fasting or fed to study the behaviour of the dosage form of bioactive compounds and food.

It presents great versatility since it allows reproduction of the physiological conditions of human digestion (digestive secretions, pH and gastric and intestinal transits, etc.) for different population groups (children, adults, elderly,...) and certain pathological situations (obesity, etc.), as well as reproducing the gastrointestinal and colonic conditions of other monogastric animals (pig,...).

The intestinal absorption of the digested food can be carried out by using intestinal cell lines in combination with the colonic fermentation system.

It allows maintenance of the anaerobic conditions required in the fermentation process, to protect the oxygen-sensitive compounds involved in the colonic fermentation.

The design allows collection of samples of the luminal content of the three sections of the colon at any time during the fermentation process of the food for analysis (metagenomic, metabolomic, SCFA,...) and to obtain information on the transformation of the food during digestion.

It incorporates a computer program for the automated control of different physiological conditions applied in colonic fermentation assays.

The characteristics of the system and the computer-control of physiological parameters open possibilities for variation of conditions that would allow the simulation of microbial dysbiosis associated to pathological conditions or due to unbalanced diets.

## 4. Study Limitations

All the data in this article come from an industrial project formed by a consortium of food sector companies. Each in vitro digestor test takes a month to complete and every metagenomic analysis represents a large part of the budget, so there are not enough resources to replicate the process more than one time for each product (food or biocompound). That is why there are not enough data to perform statistical analysis regarding the bacteria species relative abundance on the different reactors, in Figure 3.

## 5. Conclusions

Combining the in vitro colonic fermentation with 16S rRNA-based metagenomic analysis with Next Generation Sequencing (NGS) and the UHPLC-ESI-QqQ-MS/MS metabolomic analysis provides an appropriate methodology for reproducing the microbiota environment under diverse conditions (eubiosis or dysbiosis) to study the potential modulatory effects of bioactive compounds, ingredients or foods. The in vitro colonic fermentation makes it possible to reproduce the typical microbiota under dysbiosis conditions, whereas the 16S rRNA protocol allows characterization of the change in microbiota composition up to species level. In addition, the deep chemical characterization of the colonic medium allowed ascertainment of the main metabolites generated among a wide range of determined compounds. The capacity of the proposed protocol was corroborated with the studied sample. The treatment with the vegetable drink produced changes in the microbiota. Although there was no clear improvement in the diversity, the population of some bacteria increased, such as some SCFA-producing bacteria that produce propionate, acetic and butyrate. The metabolomic analysis showed an increased production of some compounds that may ameliorate health status. In conclusion, we present an interesting protocol for testing of the modulatory effects of foods or biocompounds on gut microbiota composition and metabolic activity 

## Figures and Tables

**Figure 1 foods-10-03020-f001:**
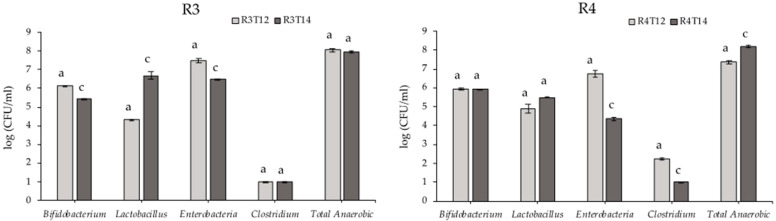
Microbiological count of the three colonic reactors: (**R3**, **R4** and **R5**) before (T12) and after the treatment (T14) (*n* = 2 ± SD). A Mann–Whitney U Test was applied for differences between two groups on a single, ordinal variable with no specific distribution. Different letters mean statistically significant differences: a, no significant difference (*p* > 0.05) and c, for a very significant difference (*p* < 0.025). Results are shown as log (CFU/mL).

**Figure 2 foods-10-03020-f002:**
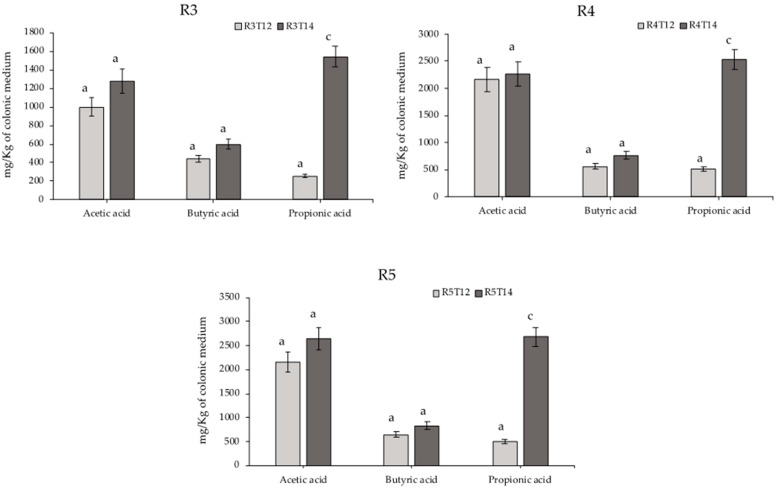
Representation of the SCFA production on the different reactors: (**R3**, **R4** and **R5**) before (T12) and after the treatment (T14) (*n* = 2 ± SD). A Mann–Whitney U Test was applied for differences between two groups on a single, ordinal variable with no specific distribution. Different letters mean statistically significant differences: a, no significant difference (*p* > 0.05) and c, a very significant difference (*p* < 0.025).

**Figure 3 foods-10-03020-f003:**
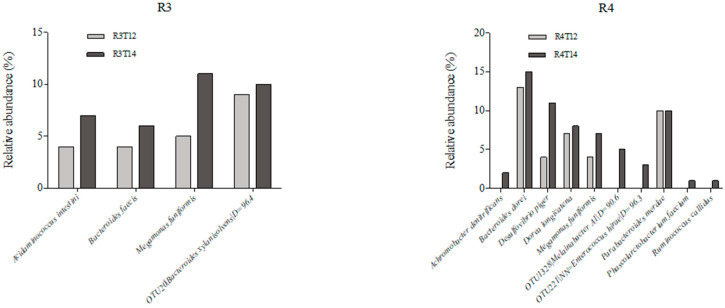
Representation of the bacteria species relative abundance on the different reactors: (**R3**) ascending colon, (**R4**) transversal colon, (**R5**) descending colon, before (T12) and after the treatment period (T14) with the biocompound.

**Figure 4 foods-10-03020-f004:**
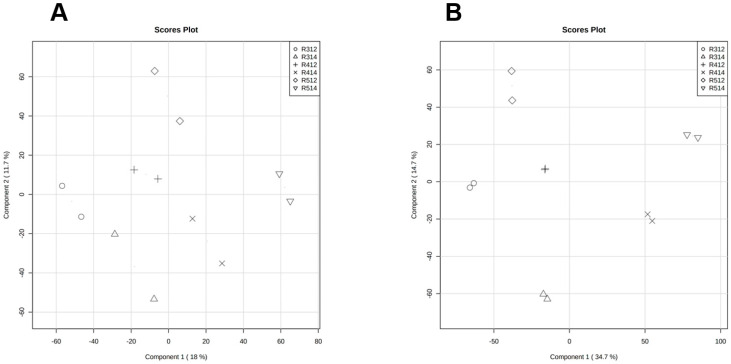
Representation the PLS-DA Score Plots obtained in the comparisons among all groups in positive (**A**) and in negative (**B**) polarity.

**Table 1 foods-10-03020-t001:** Microbiological count of the three colonic reactors during the fermentation of the vegetable drink. R3T12 represents the end of the stabilization period and R3T14 the end of the treatment period. Results are shown as log (CFU/mL).

	Days	Bifidobacterium	Lactobacillus	Enterobacteria	Clostridium	Total Anaerobic
R3T12	5	7.48 ± 0.05	6.19 ± 0.04	5.4 ± 0.01	1.00 ± 0.02	7.53 ± 0.11
	8	5.59 ± 0.03	5.5 ± 0.57	5.48 ± 0.01	1.00 ± 0.02	8.2 ± 0.07
	12	6.16 ± 0.01	4.31 ± 0.01	7.49 ± 0.12	1.00 ± 0.01	8.05 ± 0.09
R3T14	2	4.00 ± 0.02	3.2 ± 0.02	5.43 ± 0.07	1.00 ± 0.01	5.88 ± 0.11
	7	4.59 ± 0.11	6.55 ± 0.01	5.42 ± 0.04	1.00 ± 0.02	6.66 ± 0.19
	10	4.25 ± 0.02	6.66 ± 0.08	4.98 ± 0.04	1.00 ± 0.01	6.62 ± 0.23
	14	5.45 ± 0.01	6.68 ± 0.21	6.48 ± 0.01	1.00 ± 0.01	7.97 ± 0.06
R4T12	5	7.97 ± 0.08	7.79 ± 0.07	6.04 ± 0.02	4.37 ± 0.07	8.34 ± 0.12
	8	6.60 ± 0.06	6.37 ± 0.78	6.42 ± 0.08	3.25 ± 0.08	8.4 ± 0.09
	12	5.93 ± 0.05	4.87 ± 0.24	6.74 ± 0.19	2.25 ± 0.05	7.37 ± 0.08
R4T14	2	4.30 ± 0.02	3.89 ± 0.05	5.16 ± 0.12	1.93 ± 0.09	7.65 ± 0.09
	7	4.60 ± 0.13	5.85 ± 0.07	4.34 ± 0.26	1.00 ± 0.02	7.9 ± 0.07
	10	5.20 ± 0.02	6.02 ± 0.04	5.16 ± 0.05	1.00 ± 0.01	7.73 ± 0.05
	14	5.91 ± 0.03	5.48 ± 0.02	4.35 ± 0.07	1.00 ± 0.01	8.20 ± 0.05
R5T12	5	7.95 ± 0.02	7.48 ± 0.02	6.85 ± 0.02	6.46 ± 0.01	8.28 ± 0.04
	8	7.06 ± 0.02	7.14 ± 0.25	6.34 ± 0.01	5.33 ± 0.04	7.97 ± 0.02
	12	5.66 ± 0.07	5.19 ± 0.07	6.16 ± 0.02	3.88 ± 0.05	7.63 ± 0.07
R514	2	4.00 ± 0.03	4.61 ± 0.20	5.09 ± 0.03	3.46 ± 0.03	7.66 ± 0.09
	7	4.72 ± 0.09	6.22 ± 0.01	4.27 ± 0.05	2.41 ± 0.01	7.8 ± 0.07
	10	4.62 ± 0.07	5.99 ± 0.04	5.30 ± 0.66	3.06 ± 0.02	7.91 ± 0.03
	14	5.46 ± 0.02	6.16 ± 0.02	4.93 ± 0.16	2.46 ± 0.08	8.11 ± 0.05

**Table 2 foods-10-03020-t002:** SCFA production in the three reactors after the stabilization period (T12) and the treatment period (T14) (mg/Kg of colonic medium).

	Acetic Acid	Butyric Acid	Propionic Acid
R3T12	999 ± 98	441 ± 37	258 ± 17
R3T14	1281 ± 127	597 ± 54	1543 ± 116
R4T12	2157 ± 226	561 ± 52	509 ± 40
R4T14	2267 ± 222	766 ± 69	2531 ± 117
R5T12	2155 ± 209	646 ± 59	509 ± 42
R5T14	2645 ± 238	835 ± 76	2684 ± 196

**Table 3 foods-10-03020-t003:** Bacterial species abundance in the three colonic sections. R3T12 represents the end of the stabilization period and R3T14 the end of the treatment period. Values represent relative abundance %.

R3	R3T12	R3T14	R5	R5T12	R5T14
*Acidaminococcus intestini*	4.45	6.96	*Acidaminococcus intestini*	2.79	3.88
*Bacteroides faecis*	3.63	5.56	*Acinetobacter septicus*	0.00	0.94
*Megamonas funiformis*	4.68	11.17	*Alistipes putredinis*	5.32	6.29
OTU20|*Bacteroides xylanisolvens*|D = 96.4	9.38	10.14	*Anaerotruncus colihominis*	3.36	4.74
R4	R4T12	R4T14	*Bacteroides dorei*	13.09	14.85
*Achromobacter denitrificans*	0.00	1.87	*Desulfovibrio piger*	4.85	10.99
*Bacteroides dorei*	12.98	14.78	*Megamonas funiformis*	3.86	6.03
*Desulfovibrio piger*	3.56	11.21	OTU1096|*Alistipes indistinctus*|D = 96	2.60	3.06
*Dorea longicatena*	7.32	7.60	OTU1235|*Parabacteroides distasonis*|D = 96.7	0.88	2.70
*Megamonas funiformis*	4.15	6.83	OTU1328|*Melainabacter* A1|D = 90.6	0.00	4.27
OTU1328|*Melainabacter* A1|D = 90.6	0.00	4.78	OTU1572|*Parabacteroides distasonis*|D = 95.3	0.88	3.47
OTU221|*Enterococcus hirae*|D = 96.3	0.00	2.65	OTU577|*Megasphaera elsdenii*|D = 96	0.00	1.91
*Parabacteroides merdae*	9.95	10.03	OTU80|*Eubacterium desmolans*|D = 94.1	5.76	6.82
*Phascolarctobacterium faecium*	0.00	0.91	OTU82|*Clostridium lactatifermentans*|D = 93.1	0.00	4.47
*Ruminococcus callidus*	0.00	0.91	*Phascolarctobacterium faecium*	2.39	7.08
			*Phascolarctobacterium succinatutens*	4.29	5.85

**Table 4 foods-10-03020-t004:** Bacterial genus abundance in the three colonic sections. R3T12 represents the end of the stabilization period and R3T14 the end of the treatment period. Values show relative abundance %.

Tax	Inoculation	R3T12	R3T14	R4T12	R4T14	R5T12	R5T14
*Acidaminococcus*	0.00	4.45	6.96 ^	3.03	2.65 *	3.24	3.88 ^
*Acinetobacter*	0.00	0.00	0.00	0.00	0.00	0.00	0.94 ^
*Anaerotruncus*	2.22	0.00	0.00	3.03	0.00 *	3.36	4.74 ^
*Bacteroides*	12.83	15.16	10.62 *	14.90	14.98 ^	14.79	14.94 ^
*Chryseobacterium*	0.00	0.00	6.77 ^	0.00	0.00	0.00	0.00
*Cloacibacillus*	0.00	0.00	0.00	0.00	0.91 ^	0.00	3.47 ^
*Desulfovibrio*	9.04	0.00	0.00	3.56	11.21 ^	4.85	10.99 ^
*Dorea*	8.05	0.00	0.00	7.32	7.60 ^	8.63	6.64 *
*Enterobacter*	0.00	7.07	10.03 ^	4.64	4.95 ^	4.41	5.30 ^
*Enterococcus*	0.00	3.10	0.00 *	0.00	3.16 ^	0.00	0.94 ^
*Lachnoclostridium*	11.62	1.54	0.00 *	10.65	10.72 ^	11.07	11.69 ^
*Lachnospira*	10.64	0.00	4.99 ^	7.65	0.91 *	8.38	3.59 *
*Lysinibacillus*	0.00	11.20	11.20 ^	6.63	5.78 *	2.79	7.06 ^
*Megamonas*	0.00	4.68	11.23 ^	4.15	6.85 ^	3.86	6.07 ^
*Megasphaera*	0.00	5.86	8.20 ^	4.05	3.74 *	3.48	4.52 ^
*[Melainabacter]*	7.09	0.00	0.00	1.14	10.04 ^	0.00	9.24 ^
*Olsenella*	2.22	0.00	0.00	0.00	0.00	0.00	1.91 ^
*Parabacteroides*	9.59	0.00	0.00	9.95	10.03 ^	10.42	8.67 *
*Parasutterella*	4.99	0.00	0.00	1.14	3.74 ^	2.60	4.40 ^
*Phascolarctobacterium*	8.22	0.00	0.00	0.00	0.91 ^	4.57	7.59 ^
*Propionibacterium*	0.00	0.00	0.00	0.00	0.00	0.00	1.50 ^
*Tyzzerella*	6.89	0.00	0.00	5.84	0.00 *	5.19	6.32 ^

^: Bacterial genus increase; *: Bacterial genus decrease.

**Table 5 foods-10-03020-t005:** Bacterial diversity numeric classification of the inoculum and the three reactors. T = 12 represents the end of the stabilization period and T = 14 the end of the treatment period.

SW Div	Inoculation	R3	R4	R5
T = 12	3.87	2.03	3.51	3.55
T = 14		2.10	2.61	3.02

**Table 6 foods-10-03020-t006:** Some metabolites whose levels changed in the three reactors.

m/z	Tr: (min)	Reactor (Polarity)	Intensity T = 12 (Days)	Intensity T = 14 (Days)	Putative Metabolites
415.1389	5.64	R3 (neg)	4.19	6.14	Heptamethoxyflavanone/Eleganin
367.1145	5.75	R3 (neg)	4.19	5.93	Barpisoflavone/Glisoflavone
401.3422	5.81	R4 (neg)	3.27	5.68	Hydroxy-dihydrovitamin D3
517.4152	7.22	R5 (neg)	3.54	5.62	Carotene/Lycopene
104.1036	2.82	R4 (pos)	5.60	7.13	Choline

Tr: retention time; intensity expressed on a logarithmic scale.

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
