# Peer review of "An In Vitro Protocol to Study the Modulatory Effects of a Food or Biocompound on Human Gut Microbiome and Metabolome"

_foods, 2021, doi:10.3390/foods10123020_

Round 1
Reviewer 1 Report
The authors answered the reviewers' comments and questions convincingly, and therefore.
The manuscript entitled “In vitro protocol to study the modulatory effects of a food or biocompound on human gut microbiome and metabolome” is a methodology paper that is suitable for the special issue “New Methods and Technology Used in the Characterization of Different Beverages from Different Types of Food Sources” by Carles Rosés i Pol et al.
During the last decade, it became apparent that In Vitro Simulation of Human Colonic Fermentation models that mimic the colon digestive system are essential for the elucidation of the effects of different food constituents on gut microbiota and its metabolites. Several models have been developed so far and were the subject of a recent review (Veintimilla-Gozalbo et al. Applied Sciences 11, no. 17: 8135. https://doi.org/10.3390/app11178135). The current study adds a new methodology, with certain advantages over the existing protocols as described in the revised manuscript (lines 313-336).
Author Response
Authors will ask for the English language and style revision provided by the journal.
Reviewer 2 Report
In this manuscript, Roses et al describe an in vitro protocol simulating the physiological conditions of the digestive system to study the effects of foods/biocompounds on gut microbiome and metabolome.
Overall the manuscript is interesting and clear, even if to improve the manuscript, this reviewer suggests:
-to amplify introduction section in order to better understand the state of art of their study enriching it of more references;
- The authors should provide SEM value for each data reported into the graph, improving figure 1, 2 and 3.
Author Response
We would like to ask the journal for an English language and style revision. The other changes are explained in the file submitted.

Reviewer 3 Report
This manuscript is very interesting and fits the theme of the Foods journal.
However, in my opinion, the Authors must clearly formulate the aim of this study described in this manuscript and give it into the Introduction.
I have comments to the following sections:
- "2.1. In vitro Digestion Models" - In my opinion, this point should present and discuss the model used in these studies, not the advantages and disadvantages of models described in the literature (there is a place in the discussion of results or in the Introduction).
- Your model includes Dynamic Gastrointestinal and Colonic Fermentation Stages, and what about saliva? The saliva contains digestive enzymes for carbohydrates, important for bacteria. In my opinion, this should be included in the discussion and the Introduction.
- “2.2. Description of the Dynamic gastrointestinal and colonic fermentation model” - What was the initial pH fixed for gastric digestion?
- “2.3. Faecal inoculum” - At what values pH was maintained?
- “2.4. Process description and duration” - Please correct "CFU/ML 'instead of" CUF/ML". What do you mean using the term "colonic media"? I want to note that the MRS Agar medium is used for lactobacilli, but not to lactic acid bacteria in general. Please correct it in a manuscript. Please enter the medium incubation conditions for each bacterial group separately.
- “2.6. Short fatty acid analyses” - Please enter information about internal standard as well as analysis parameters.
- “2.7. DNA Extraction”, “2.8. Metagenomic Data: Library Preparation”, & “2.10. Richness and Evenness” - Please provide references for these analyzes.
- And what about statistical analysis? Please enter how many repetition experiments have been made in how many individual analyzes have been made? What statistical analysis was made?
- The presented scope of results does not coincide with the methodologies described in the manuscript. Please enter exactly the results obtained from the methods described in the methodology of this manuscript.
- Table 1 and Table 2 – They present very interesting data, but not subject to statistical analysis, why?
- Figures 1, 2, and 3 - Please specify the explanation for the legend as well as the value of standard deviation or statistical analysis parameters.
- The discussion of the results is very limited, I suggest the authors to expand it.
- “4. Conclusions” - This section is not consistent with the presentations and methodology described in the manuscript. The more so because you did not give the target of the research, and the conclusions must respond to the purpose of research and summarize the described research methods.
Author Response
We would like to ask the journal for an English language and style revision. The other changes are explained in the submitted file.

Round 2
Reviewer 3 Report
I can see that the authors tried to correct the manuscript taking into account the suggestions of the reviewers. And they almost succeeded :) I only notice minor bugs now that still need to be corrected:
- Line 142: Bifidobacterium, Enterobacteriaceae
- Line 307: Not all Enterobacteria are potentially pathogenic species. I suggest that you write: " Furthermore, bacteria from the Enterobacteriaceae family, including potentially pathogenic bacteria, show ....."
- Line 312: “Propionic acid level showed a very significant….”
Author Response
- Line 142: Bifidobacterium, Enterobacteriaceae. Correction done.
- Line 307: Not all Enterobacteria are potentially pathogenic species. I suggest that you write: " Furthermore, bacteria from the Enterobacteriaceae family, including potentially pathogenic bacteria, show ....." Correction done.
- Line 312: “Propionic acid level showed a very significant….”
- Correction done.
About the English language and style are fine/minor spell check required, we would like the journal to proceed with the English revision system. We are waiting to receive the payment of the invoice generated!
This manuscript is a resubmission of an earlier submission. The following is a list of the peer review reports and author responses from that submission.
Round 1
Reviewer 1 Report
The manuscript by Roses et al. is an appealing and helpful methodology paper on a newly developed dynamic gastrointestinal and colonic fermentation model. The report is straightforward and comprehensive, with substantial details of the methodology used. Continuous multistage dynamic fermentation models are valuable for evaluating spatial and temporal changes of the colonic microbiota composition in response to dietary ingredients.
However, several models have been developed so far. What is lacking from the manuscript is the pros and cons of their newly reported model and a direct comparison with other existing models.
Author Response
We would like to thank the reviewers for their thoughtful comments and suggestions for improving this manuscript about a dynamic gastrointestinal and colonic fermentation model. We have taken into account the comments and tasks proposed by the reviewers, and we believe that in addressing these comments, this reviewed manuscript is considerably improved. A detailed point-by-point response to the reviewer’s comments can be found after each question in the document attached. Changes in the manuscript have been marked up using the “Track Changes” function from MS Word, so any changes can be easily viewed by the editors and reviewers.

Reviewer 2 Report
This study reported a mimic colon digestive system to evaluate the effects of various food components which utilized by gut microbiota. The present study offered some evidences of simulated status of colon digestive processes between gut microbiota and food components, however, I do have some questions raised.
- Please discuss more basic information regarding the mimic colon digestive system in Introduction section, especially the interactions of dysbiosis status.
- There were many physical factors that affects the production and composition of gut-related metabolites. If authors do not focus on the such interactions (i.e. absorption and gut leak), there is a black box to mimic the interactions of target dietary components and microbiomes.
- For the transition time, temperature, pH, ionic status in guts, such information should be provided to clarify the possible interactions involving between gut microbiota and special food components. Under this design, some basic gut biochemical parameters (secondary bile acids or related tight junction proteins in guts) have been omitted, such limitations should be provided in text.
- Over all, this was an interesting design but lack of novelty and direct evidences to demonstrate the authors’ aim. Authors should provide the solid data to convince reader that mimic apparatus is a powerful unit to provide the information of food components and gut microbiota.
Author Response

(The authors gave the same response as above.)
